# The Effect of Mineral Wool Fiber Additive on Several Mechanical Properties and Thermal Conductivity in Geopolymer Binder

**DOI:** 10.3390/ma17020483

**Published:** 2024-01-19

**Authors:** Beata Łaźniewska-Piekarczyk, Dominik Smyczek

**Affiliations:** 1Department of Building Processes and Building Physics, Faculty of Civil Engineering, The Silesian University of Technology, Akademicka 5, 44-100 Gliwice, Poland; dominik.smyczek@saint-gobain.com; 2Saint Gobain Construction Products Polska Sp. z o.o., Okrężna 16, 44-100 Gliwice, Poland

**Keywords:** geopolymer, mineral wool waste, mechanical properties, thermal conductivity coefficient, mineral wool pulverization

## Abstract

The article discusses the effect of additives of waste mineral wool fibers on geopolymer binder. This is an important study in terms of the possibility of recycling mineral wool waste. The paper describes an effective method for pulverizing the wool and the methodology for forming geopolymer samples, labeled G1 for glass-wool-based geopolymer and G2 for stone-wool-based geopolymer. The compressive and flexural strengths and thermal conductivity coefficient of the geopolymer with the addition of mineral fibers were determined. The key element of the article is to verify whether the addition of mineral wool fibers positively affects the properties of the geopolymer. The results obtained prove that the addition of fibers significantly improves the flexural strength. For the G1 formulation, the ratio of compressive strength to flexural strength is 18.7%. However, for G2 samples, an even better ratio of compressive strength to flexural strength values of 26.3% was obtained. The average thermal conductivity coefficient obtained was 1.053 W/(m·K) for the G1 series samples and 0.953 W/(m·K) for the G2 series samples. The conclusions obtained show a correlation between the porosity and compressive strength and thermal conductivity coefficient. The higher the porosity, the better the thermal insulation of the material and the weaker the compressive strength.

## 1. Introduction

The article addresses the effect of the addition of mineral fibers, specifically waste insulation wool fibers, on the properties of the geopolymer binder. The parameters studied are the compressive and flexural strength and thermal conductivity coefficient. This work aims to determine what effect the addition of mineral fiber has on the insulating and mechanical properties. The study is relevant to potential areas of application for mineral-wool-based geopolymers. The geopolymer mortar additive selected by the project team is mineral wool waste. In the present age of thermal modernization, the production volumes of this material are increasing, which is related to the growing amount of waste generated, and for this reason, this material has been selected [1].

The available literature references describe the properties of geopolymers primarily based on fly ash or metakaolin. Nevertheless, there are few publications describing the properties of mineral-wool-based geopolymer. These publications only describe the compressive strength properties, with no information on flexural strength. The previous publication issued by the team [2] contains both compressive and flexural strength studies; nevertheless, the research described is on metakaolin-wool-based geopolymer. The article in question presents research on geopolymers made of wool and norm sand. Publications on the thermal conductivity coefficient also only deal with its properties for metakaolin-based geopolymers. One publication describing the thermal conductivity coefficient of mineral-wool-based foamed geopolymers is different in methodology.

Due to its specific physical and chemical properties, mineral wool is a type of waste that is problematic to manage. It is even considered as a non-recyclable waste [3]. In particular, mineral wool is non-combustible, so it is not suitable as a raw material for alternative fuels [4]. The mechanical processing of this waste is also a challenge. Mineral fibers, despite their low density, are very abrasive, which causes abrasion to shredding equipment [5]. Moreover, the large volume relative to the low mass makes it economically unjustifiable to transport this waste over long distances. Due to the specific characteristics of the waste, it is stored in industrial landfills, contributing to environmental pollution [6]. Unfortunately, mineral wool also causes problems in landfills, causing lack-of-landfill instability [7]. For this reason, they require special landfilling conditions. The literature sources distinguish between two sources of mineral wool waste: the stream coming from manufacturing plants and that coming from demolition. Often, the waste coming from manufacturing plants is internally captured by factories and sent for internal recycling. The waste generated in the factory during production is a waste that is simpler to manage because it is not contaminated with other building materials, so its chemical composition is preserved [1]. The second stream of wool waste comes from building demolition and the decommissioning of agricultural crops, and poses a much more significant issue. This type of waste often pollutes with other construction waste. The problem involves its altered shape, form, and numerous contaminants [8]. An additional aspect is the moisture that is absorbed by the wool. All the above issues disqualify it from reuse in production processes due to the quality of the product [9]. The European Union waste catalog classification also separates these two waste streams. Wool demolition waste is in Group 17 (waste from the construction, repair, and dismantling of construction and road infrastructure (including soil and earth from contaminated sites)). On the other hand, wool waste from manufacturing plants is classified as Group 10 (waste from thermal processes) [10].

The above characteristics compound the fact that recycling mineral wool waste is a challenge in current environmental engineering. Mineral wool waste is used in industries including ceramics as an additive to precast ceramic products [11], concrete mortar to improve its mechanical properties [12,13], composites [14], or briquettes [15]. One contemporary form of mineral wool recycling reported in the published literature sources is immobilization in geopolymers [16]. Preliminary studies demonstrate that mineral fibers, due to their aluminum and silicon content, can be an excellent raw material for geopolymer production. This treatment avoids the cost of landfilling; in addition, it makes it possible to generate additional income from the sale of the construction material, which balances the high cost of processing the waste. Also, the available publications describe how geopolymers have many pro-environmental properties. Among them is the ability to immobilize phenol and formaldehyde emissions from mineral wool [17]. Geopolymers represent a group of synthetic, inorganic, aluminosilicate polymers. Although these materials are currently experiencing a renaissance due to their great potential in waste recycling, according to the literature, geopolymers were used as early as 25,000 years ago [2]. The definition of geopolymers was introduced into the literature by Professor Joseph Davidovits in 1978 [18]. The reaction behind the formation of this material is the synthesis of silicon and aluminum. The chemical composition of a geopolymer is comparable to zeolite, while geopolymers have an amorphous inner structure [19]. The basis for the formation of a geopolymer is a bulk material whose chemical composition is dominated by aluminosilicates. The second important element in the composition of a geopolymer is an activator, which is the chemical mixture designed to create an alkaline environment in which the bonding reaction will take place. There are numerous publications describing the characteristics of geopolymers based on metakaolin [20], fly ash [21], or blast furnace slag [22]. The literature references state that mineral wool can be a precursor for geopolymer production, while there are limited publications on their properties. This article addresses this research gap.

## 2. Materials and Methods

### 2.1. Materials

#### 2.1.1. Mineral Stone and Glass Wool

The materials used to produce the samples of geopolymers were stone wool (Figure 1a) and glass wool (Figure 1b), which are waste products from the production process of insulation wool.

According to the current European-Union-wide waste catalog in the Regulation [23], the stream of mineral wool waste from production streams is separated from demolition waste. Code 10 11 03 is used for glass wool waste and code 10 12 99 for stone wool waste; these are listed under production waste, and code 17 06 04 is used for mineral wool in general as a demolition insulation material. After years of wool use and contamination by other building materials, it is often difficult to qualify the type of wool without specialized testing, hence the general classification. The wastes mentioned are fundamentally different from each other. Wool plants use segregation and developed internal recycling methods [24]. This results in the acquired waste being dry, homogeneous, and not contaminated with other construction waste as is the case with demolition waste. Equally, there are important differences between glass wool and stone wool. The raw materials for stone wool are gabbro, basalt, slag, dolomite, bauxite, and briquette from stone wool waste. It is melted most often in a cupola, using coke and gas. Stone wool usually has a darker color, is denser and stiffer, and has a resistance to higher temperatures of 1050 °C. These properties mean it is most often used in flat roofs and for technical applications such as high-temperature installation insulation. As opposed, the raw materials for glass wool are sand, soda, feldspar, lime meal, dolomite meal, and internal or external glass cullet. Glass wool is lighter, with a yellowish color. It has a lower density, which results in a better thermal conductivity coefficient. These features make it an ideal material for housing construction, especially for pitched roof insulation [25].

The wool used in the study is polymerized and dry, in the form of granules with a fiber length of 3–5 cm for stone wool and 5–10 cm for glass wool. In order to achieve a homogeneous bulk material, the wool samples were pulverized. After analyzing the available literature [26] on the pulverizing of mineral wool fibers, a ball mill of the Los Angeles type complying with EN 1097-2 [27] was selected to grind the wool (Figure 2).

To obtain a homogeneous powder with the specifications presented in Table 1, the mill setting was 3000 rpm for both samples. The grinding of glass wool with the addition of electrocorundum resulted in a powder labeled WS2 (WS2), while the grinding of stone wool resulted in a powder labeled WM2 (WM2). The material obtained from grinding was fine-grained, with a grayish, light-yellow color. The WS2 had an average grain size of 18.32 μm. In comparison, the second geopolymer meal used for the G2 recipe (G2), WM2, had an average grain size of 21.90 μm. Although similar in grain size, the ground stone wool had a much lower specific surface area of 0.92 m^2^/g, compared to glass wool (WS2) with a specific surface area of 2.36 m^2^/g.

The significant difference in specific surface area between WS2 and WM2 despite similar grain size was a secondary research problem. Publications on studies of soils and the correlation between the content of organic parts and the specific surface area were helpful [28,29]. The described research results demonstrate the correlation: the more organic parts in the studied material, the smaller its specific surface area. Stone wool, due to its additional technical applications, contains more resin, which translates into more organic parts. Furthermore, the mineral wool milling process shortens the fibers, while maintaining their thickness. Glass fibers are thinner and more fragile. For this reason, they break into smaller particles. The second important aspect related to the milling process is the proportion of electrocorundum. In order to achieve a suitable geopolymerization process, the geopolymer powder should be rich in aluminum and silica. The chemical analysis of pulverized glass wool without electrocorundum (WS1) and pulverized stone wool without electrocorundum (WM1), presented in Table 2, shows the proportion of aluminum and silicon without enrichment. It can be concluded that, especially in the case of glass wool, silicon is predominant, with a small amount of aluminum. The values are not balanced.

For this reason, the project team decided to add electrocorundum to the grinding process (Figure 3). This procedure enabled us to enrich the geopolymer powder with aluminum and to improve the grinding process. Electrocorundum is chemically an aluminum oxide. It is a synthetic type of the naturally existing mineral corundum. It is a very abrasive and hard mineral with a Mohs hardness of 9. Electrocorundum is obtained synthetically by melting bauxite in a resistance-arc furnaces. In the present study, white noble electrocorundum was used. It has a more homogeneous chemical composition (higher Al_2_O_3_ level) compared to ordinary brown electro-corundum. It is widely used in the abrasive industry. It is low-cost because it is a synthetic variety [30].

Based on the team’s previous work, it was found that the addition of corundum to the grinding process significantly affects the grinding time of wool. The literature sources indicate what applications the geopolymer can have, depending on the Al:Si ratio [31]. For common use and the purposes of this work, a suitable Al:Si ratio of 1:1 was assumed. The parameters of the milled material after enrichment are shown in Table 3.

With the addition of electrocorundum to the mill feed, there is a noticeable difference in the proportions of aluminum and silicon in the WS2 and WM2 geopolymer powders under test. Both glass and stone wool significantly increased the proportion of aluminum. As such, the prepared geopolymer powder (Figure 4) was used to produce the geopolymer used in further studies.

#### 2.1.2. Mineral-Wool-Based Geopolymers and Production Methodology

Geopolymer powders, described above, were used to prepare samples of the tested geopolymers. The bulk material, rich in aluminum and silicon, was mixed dry in a premix with GRACE brand plasticizer art no. 1695 Acosal Fluid 307 (PSR) (W. R. Grace & Co.-Conn., Columbia, MD, USA) [32] and norm sand (NSD). A chemical activator (CHA) was then added to the bulk material, specifically, a mixture of an 8 M solution of NaOH and sodium water glass with a molar ratio of 3.0 and a density of 1.41 g/cm^3^ was used. The ratio of water glass to NaOH was 2.5. To make the samples, 450 g of geopolymer powder (WS2 and WM2), 1350 NSD, and 25 g of PSR were used. For the G1 formulation, 483.18 g of CHA was added, whereas for the G2 formulation, 331.56 g was added. For the strength test, the geopolymer was formed into 20 × 20 × 160 mm size blocks (Figure 5) and heat-treated for 48 h at 70 °C. The samples for the thermal conductivity coefficient test (Figure 6) were 60 mm-diameter, 10 mm-thick disk-shaped samples aged under the same conditions. After heating at 70 °C, the tested geopolymers were cured under laboratory conditions for 26 days under air conditions of ambient temperature, about 20 °C, and humidity of about 50%.

After the ageing time, the samples were unmolded. The geopolymer samples were evaluated, and their density and porosity were tested. Samples of the geopolymer based on WS2 were labeled with formula number G1, while the geopolymers based on WM2 were labeled G2. The results are shown in Table 4.

The tested samples have similar densities, oscillating in the range of 2.464–2.592 g/cm^3^. G2 samples are characterized by a higher porosity of 22.61% compared to G1 samples with a porosity of 20.91%. Similarly, the volume of pores found inside the sample is larger for G2 samples, amounting to 0.1127 cm^3^/g.

### 2.2. Research Methods

The research procedure involved conducting strength tests to determine compressive and flexural strengths, as well as investigating the thermal conductivity coefficient of mineral-wool-based geopolymers. The research was intended to evaluate the effect of mineral fiber additives to the geopolymer formulation. The geopolymer samples used for the study are described in Section 2.1.2.

#### 2.2.1. Mechanical Properties

Compressive strength testing assesses the properties of a geopolymer overall. This attribute depends on many factors, such as the proportion of ingredients used and the method of manufacture. The flexural and compressive strength of the wool-based geopolymer binders and mortars were tested according to EN 196-1:2016 [33]. A Controls model 65-L27C12 testing apparatus was used for the tests. The tests were performed after 28 days of maturation under air conditions of about 20 °C and humidity of about 50%. In the compressive strength test, cubic specimens of 40 mm × 40 mm × 40 mm were used. In the first instance, the bending strength test was performed, then the compressive strength test. When the specimens were damaged by the force exerted by the test apparatus, the highest value was recorded.

#### 2.2.2. Thermal Conductivity

The study was crucial for the comparison of the insulating properties of mineral wool and geopolymer based on wool waste. The hypothesis investigated was whether the use of mineral wool would improve the insulating properties of the geopolymer. Measurements were taken using the stationary method in a FOX 50 apparatus (TA Instruments, New Castle, DE, USA) with heat flux sensors in a symmetrical arrangement meeting the requirements of PN-ISO 8301:1998 [34] and PN EN 12664:2002 [35], at an average test temperature of 23 °C and at dT = 10 °C. Two series of three samples of 62 mm diameter and 12 mm thickness were tested. Before measurement, the samples were ground on the top and bottom surfaces and conditioned under constant temperature and humidity conditions. 

## 3. Results

### 3.1. Compressive and Flexural Strength Test Results

The results for strength for the series of samples from the formulas labeled G1 and G2, respectively, are presented in Table 5.

The tests were carried out on a series of two samples for compressive strength testing and one sample for strength testing sequentially for each of the G1 and G2 formulas. The samples tested had dimensions equal to 40 × 40 × 160 [mm]. The average value for the compressive strength of the six samples of G1 is 31.39 MPa. The high values of flexural strength should also be noted, the average of which, from three measurements, is 5.87 MPa. For the recipe G1, the ratio of compressive strength to flexural strength values is 18.7%. This is a satisfactory result. For the G2 formulation, the average compressive strength is lower than that of the G2 formulation and is 22.56 MPa. On the other hand, the flexural strength of the G2 samples is higher than for the G1 and is 5.94 MPa. Therefore, the ratio of compressive strength to flexural strength values is even higher, equal to 26.3%. If we relate the values obtained in this study to the values available in the literature for conventional concrete and geopolymer without fiber participation, then the results obtained can be evaluated as being very satisfactory. In comparison, the average strength for geopolymers without mineral fiber additives varies widely, and for the post-activation solution, the properties are 17 MPa compressive strength and 3.7 MPa flexural strength [36]. For these values, the ratio of compressive strength to flexural strength is 21.76%. In contrast, the average ratio of compressive strength to flexural strength values for conventional concrete is up to 15% [37]. Analyzing the above figures, it is possible to extend the research brief from the previous article by the research team [17] and conclude that the fibrous structure of wool improves the compressive and flexural strengths of geopolymer mortar under alkaline activation. Also, the study in this paper showed that the ratio of flexural strength to compressive strength can be higher than 15%.

### 3.2. Thermal Conductivity

The results for strength for the series of samples from the formulas labeled G1 and G2 sequentially, along with a comparison with other results available in the literature, are presented in Table 6.

Table 6 presents the thermal conductivity coefficient values obtained using the selected building materials. Analyzing the test results, it can be concluded with a high degree of probability that the geopolymer based on wool waste has a significantly worse thermal conductivity coefficient than mineral wool. Comparing the thermal insulation properties of the tested G1 and G2 samples to conventional building materials, they are better than reinforced concrete, but worse than ceramic brick. Therefore, this result can be assessed as unsatisfactory. The results indicate that the porosity has a greater effect on insulation than the mineral wool content in geopolymers. Figure 7 and Figure 8 present the microstructure of the tested geopolymers. Numerous pores are visible in both samples.

In the samples examined, a lower thermal conductivity coefficient was achieved using the sample with higher porosity. The series of samples designated G1 had an average thermal conductivity coefficient of 1.053 W/(m·K) and a designated porosity of 20.91%. In contrast, the G2 series of samples with a higher porosity of 22.61% achieved a better thermal conductivity coefficient, with an average of 0.953 W/(m·K). The differences shown are small; nevertheless, they confirm the correlation, showing the relationship: the higher the porosity, the lower the thermal conductivity coefficient. In sample G1, a network of numerous smaller pores can be seen and are connected to each other. In contrast, in sample G2, there are apparently fewer pores on the microphotograph itself, but they are wider, closed, and more oval. An additional test, the results of which are shown in the graph below [Figure 9], demonstrates the proportion of each pore size in the sample. A higher pore volume is seen in the G2 sample.

The graph above illustrates that there are more pores of larger size in the G2 sample; these accumulate air. In the case of mineral wool products, the insulation is the air that sits between the densely packed mineral fibers. In the case of the geopolymer, its density is much higher, which results in fewer pores and, thus, less air.

## 4. Discussion

The results of the tests presented in Table 5 on the compressive and flexural strengths of the tested glass-wool-based and stone-wool-based geopolymer mortars showed their properties and the differences between them. The G1 recipe based on glass wool had a higher compressive strength than the G2 samples with comparable flexural strength. The strength of the geopolymer is influenced by many factors related to the fraction, the chemical composition of the geopolymer powder, its Al-Si ratio, the composition and amount of the chemical activator or, finally, the time and temperature of aging of the sample. There is, naturally, the question of what causes the differences in the strengths between the G1 and G2 series samples. Since the aging time and manufacturing methodology were the same, the answer must be sought in the formulation used. First of all, there are the differences between the geopolymer flours, WS2 and WM2, that were used. Despite the similar granularity, the WS2 meal used in the recipe has a much larger specific surface area. From the perspective of studying grain size distribution via laser diffraction in the case of mineral fibers, attention should be paid not only to the fiber length, which depends on the milling process, but also to the fiber thickness, which is invariant to the milling process. Glass fibers are thinner and more fragile; for this reason, the specific surface area of glass-wool-based WS2 meal is higher. In the microphotographs of the WS2 (Figure 10) and WM2 (Figure 11) powdered meal samples, the difference in the microstructure is evident. The area shown in the microphotograph of the WS2 sample shows more particles of smaller size. The WS2 sample is also more homogeneous, with no visible lava clumps during the pulping process (particles of 73 μm and 200.21 μm in the 100 μm image shown in Figure 10). Both samples have elongated particle forms. The grinding process evidently shortened the length while maintaining the thickness of the fibers. The presence of elongated particle forms has a positive effect on the flexural strength of both samples.

The better homogeneity, longer fiber length and, most importantly, higher specific surface area resulted in higher compressive strength in the G1-labeled geopolymer samples. The results for the mechanical properties of the geopolymer summarized in Table 1 prove that the addition of mineral fibers improves the flexural strength of the geopolymer for both the G1 and G2 series tested. This advantage makes them suitable for application in many industries, such as surface reclamation binders [38].

The results for the thermal conductivity coefficient presented in Table 6 make it possible to compare the tested samples with other building materials available in the literature. Despite the fact that mineral wool is the most widely used insulation material in the world, with excellent insulating properties [25], its addition did not significantly affect the thermal conductivity coefficient of the tested geopolymers. If we contrast the less-porous and more-porous materials in Table 6, it is observable that the thermal conductivity coefficient is correlated with porosity. Likewise, in mineral wool, the insulator is the air between the formed stone or glass fibers. Nevertheless, according to available publications, it is possible to obtain a thermal conductivity coefficient in a geopolymer that is close to that of mineral wool [42]. However, this requires changing the geopolymer production technology and taking the direction of producing foamed geopolymers. Available publications describe them as low-density materials with good fireproofing and insulating properties, mostly at the expense of mechanical properties [43,44,45]. Moreover, foamed geopolymers can also be obtained using waste mineral wool [40]. Another significant part of the research work presented in the article was the pulverization of the mineral wool waste. The project team made a number of observations that are scientifically relevant. Analyzing the available literature, it can be concluded that the degree of grinding of wool depends on the grinding method used [26]. The Los Angeles ball mill effectively combines a high degree of comminution and high efficiency on a laboratory scale. Analyzing the grinding process for the purpose of this article, it can be observed that the degree of fineness of wool is also affected by a longer grinding time, the addition of electrocorundum, the length of wool fibers in the charge, and a higher ratio of balls in the mill to the amount of ground material.

## 5. Conclusions

Given the unambiguous direction of the shift set by the European Union toward a closed-loop economy, geopolymers can be part of this transformation. Under their guise lies the opportunity for conventional building materials based on primary raw materials to be replaced by modern materials made from recycled materials. Nevertheless, geopolymers need to be tested to make this happen. This article describes the effect that the addition of waste mineral wool fibers has on geopolymer properties. The results obtained show some relationships and correlations. Studies of the thermal conductivity coefficient show that its value depends more on the porosity of the material than on the use of shredded mineral wool. The results show a correlation with the principle that the greater the porosity and the greater the volume of the material ports, the lower the thermal conductivity coefficient. An inverse proportionality was shown in the flexural strength test, where samples with a higher porosity had lower compressive strength. In addition, a study of the mechanical properties of geopolymers with the addition of mineral fibers confirmed the hypothesis of a beneficial effect of fiber addition on flexural strength.

The conclusions presented in this article require further research, which will be carried out by the research team. The authors plan to formulate a geopolymer with insulating properties similar to the values obtained for porous concrete; nevertheless, this requires a change in the technology of obtaining geopolymers in the direction of foamed geopolymers.

## Figures and Tables

**Figure 1 materials-17-00483-f001:**
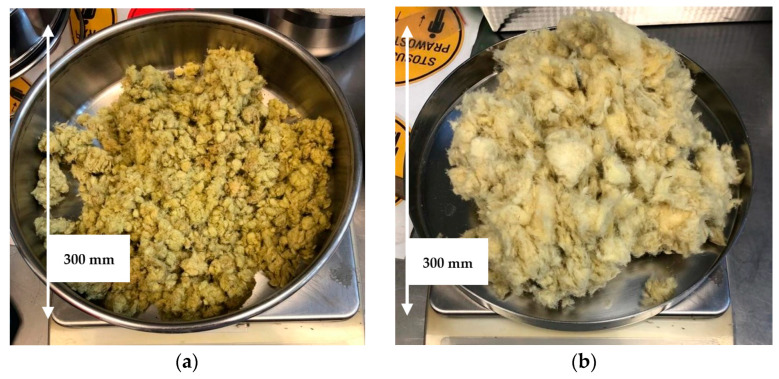
The shredded mineral wool waste used for further research on its grinding: (**a**) stone wool waste; (**b**) glass wool waste.

**Figure 2 materials-17-00483-f002:**
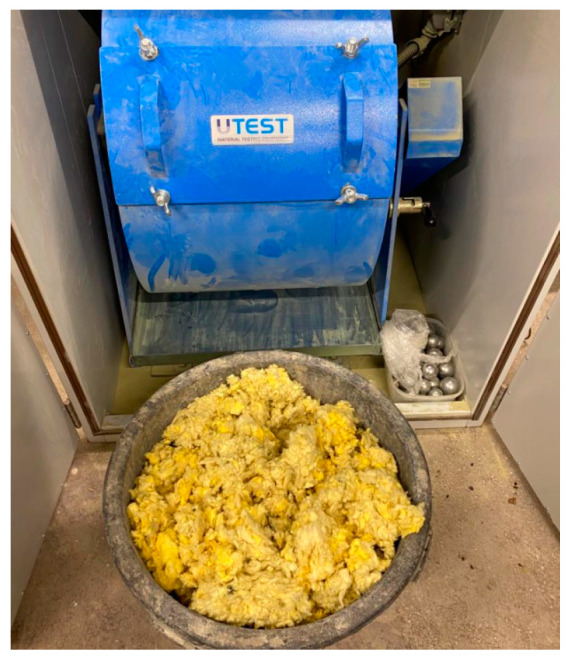
Los Angeles-type mill used to grind wool.

**Figure 3 materials-17-00483-f003:**
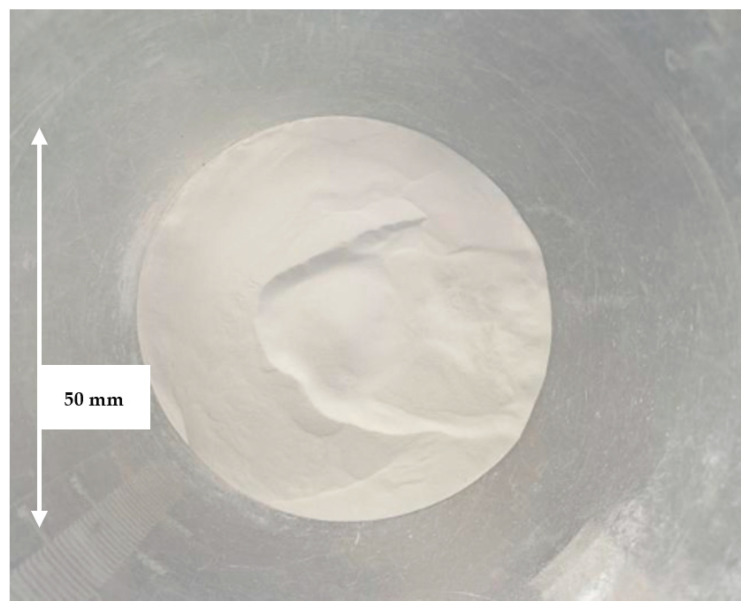
Electrocorundum used for wool grinding.

**Figure 4 materials-17-00483-f004:**
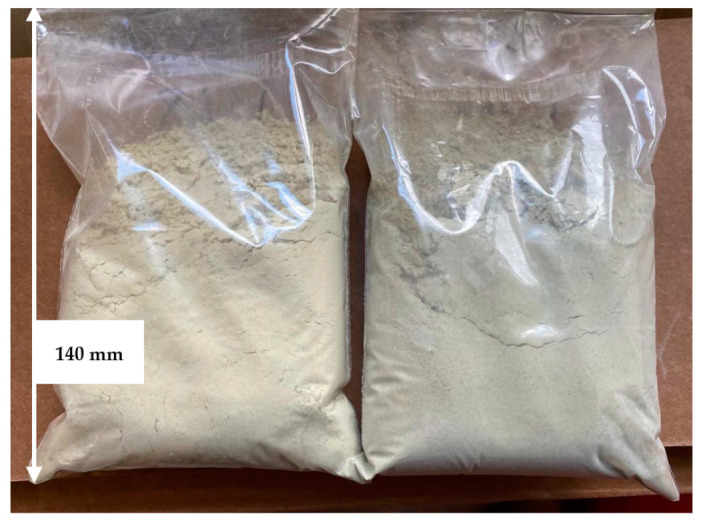
Pulverized wool with Al_2_O_3_ used for further testing. WS2 on the left, WM2 on the right.

**Figure 5 materials-17-00483-f005:**
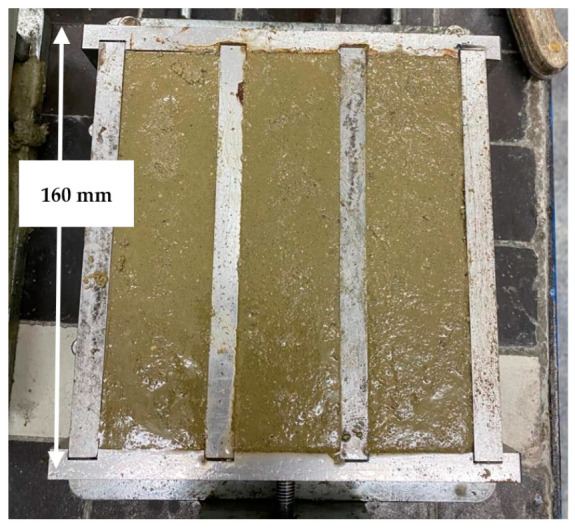
Preparing of samples for strength testing.

**Figure 6 materials-17-00483-f006:**
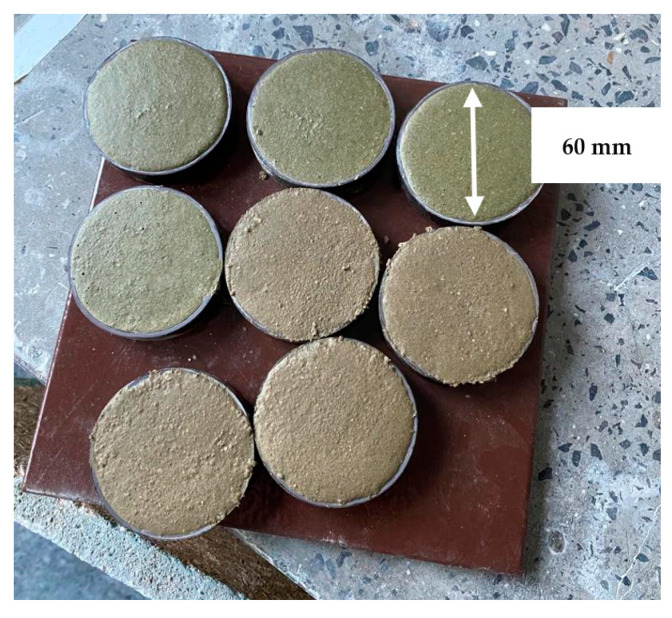
Samples for testing the thermal conductivity coefficient.

**Figure 7 materials-17-00483-f007:**
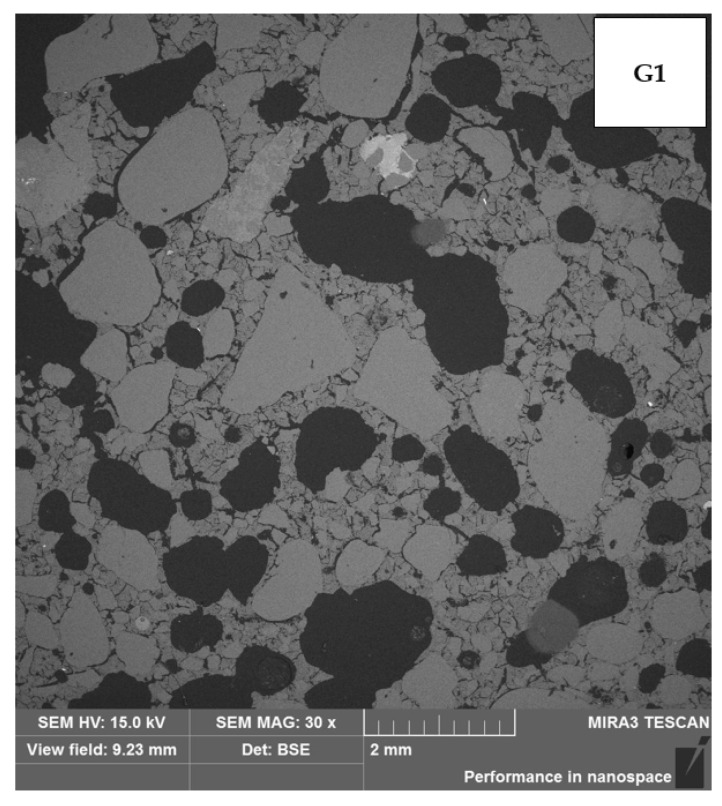
Porosity of sample G1.

**Figure 8 materials-17-00483-f008:**
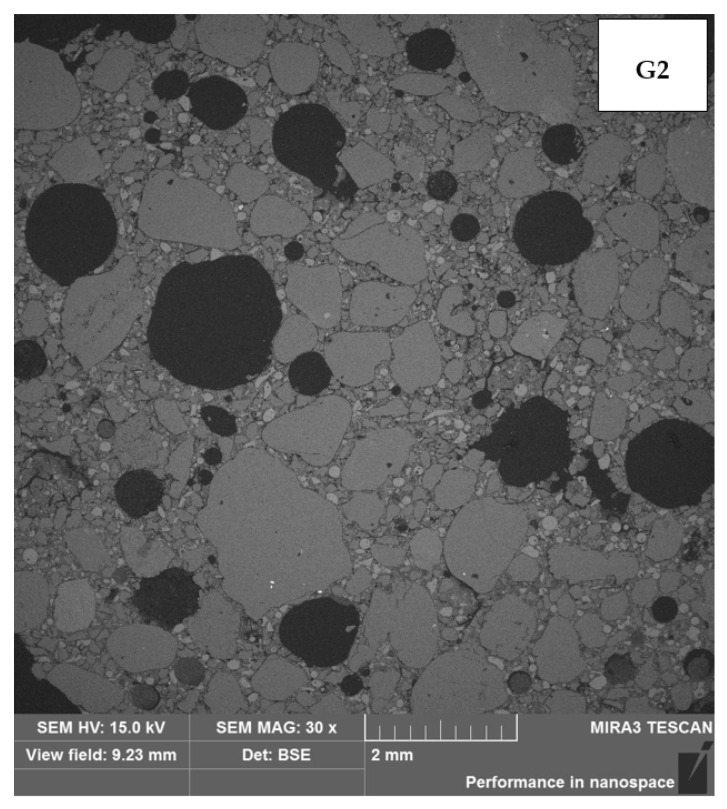
Porosity of sample G2.

**Figure 9 materials-17-00483-f009:**
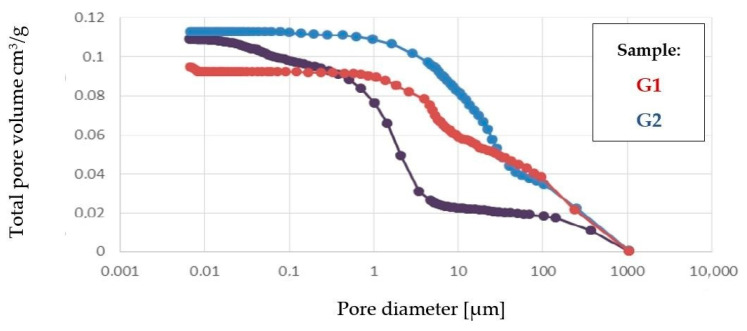
Share of individual pore sizes in the sample.

**Figure 10 materials-17-00483-f010:**
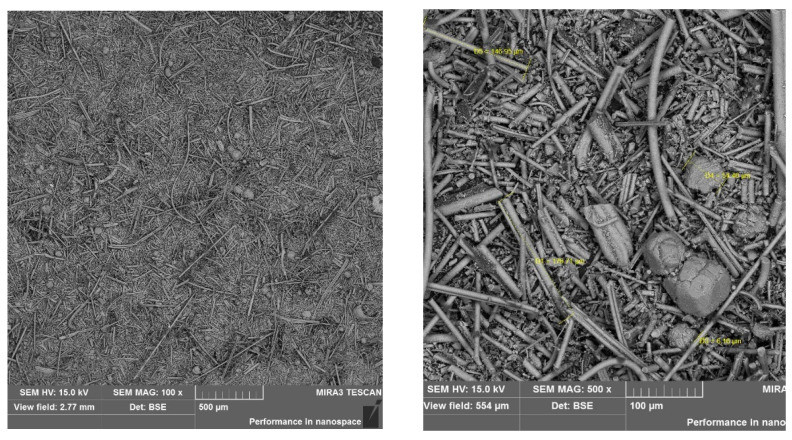
Microphotographs of sample WS2.

**Figure 11 materials-17-00483-f011:**
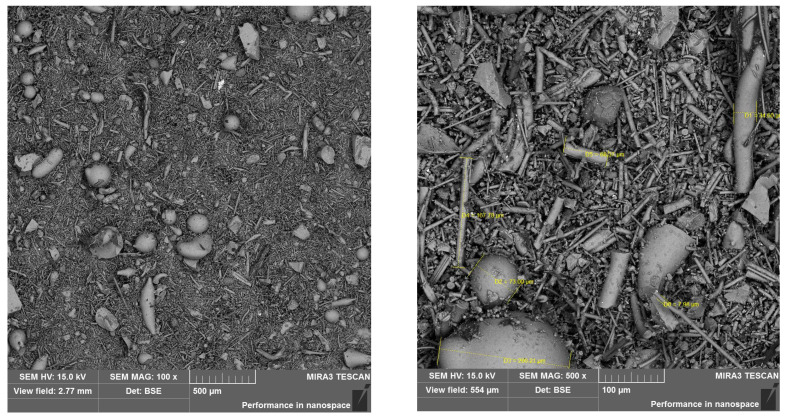
Microphotographs of sample WM2.

**Table 1 materials-17-00483-t001:** Parameters of WS2 and WM2.

Sample No.	Mode [μm]	D50 [μm]	D90 [μm]	Specific Surface Area [m^2^/g]
WS2	18.080	21.991	115.515	2.36
WM2	21.200	18.320	68.881	0.92

**Table 2 materials-17-00483-t002:** Comparison of the chemical composition of glass wool (WS1) and stone wool (WM1) without enrichment (without electrocorundum).

	O	Na	Mg	Al	Si	Ca	Ti	Cr	Fe	K
WS1	42.6	13.2	1.63	1.3	34.7	5.8	-	-	-	0.7
WM1	37.5	2	5.9	10	20.1	16.95	0.5	1	7	-

**Table 3 materials-17-00483-t003:** Comparison of chemical composition of WS2 and WM2 after enrichment (with electrocorundum; successive mass weight: WS2 25%; WM2 10%.

	O	Na	Mg	Al	Si	Ca	Ti	Cr	Fe	K
WS2	45.8	9.8	1.1	13.3	23.4	4.6	-	-	0.4	0.4
WM2	43.4	1.6	4.1	15.8	15.8	13.9	0.3	0.7	4.2	0.1

**Table 4 materials-17-00483-t004:** Comparison of the values of actual density and open porosity of the tested G1 and G2 samples.

Sample No.	Actual Density [g/cm^3^]	Porosity [%]	Open Pore Volume [cm^3^/g]
G1	2.464	20.91	0.107
G2	2.592	22.61	0.113

**Table 5 materials-17-00483-t005:** Compressive and flexural strength test results for the G1 and G2 formulations.

Sample No.	Series 1			Series 2			Series 3		
	Flexural Strength [MPa]	Compressive Strength [MPa]		Flexural Strength [MPa]	Compressive Strength [MPa]		Flexural Strength [MPa]	Compressive Strength [MPa]	
G1	6.14	32.43	31.00	6.21	31.35	30.52	5.26	31.82	31.27
G2	6.21	24.70	26.22	5.71	23.91	24.61	5.91	18.05	18.31

**Table 6 materials-17-00483-t006:** The results of the test of the coefficient of thermal conductivity of the geopolymer labeled with the G1 and G2 formulation number in comparison with other building materials.

Labeled Sample	The Thermal Conductivity Coefficient of a Single Sample, [W/(m·K)]	Average Thermal Conductivity Coefficient(Standard Deviation), [W/(m·K)]
G1	1.052	1.053 (0.006)
	1.046	
	1.061	
G2	0.951	0.953 (0.006)
	0.947	
	0.962	
Stone wool	0.037 [24]	
Metakaolin-based geopolymer	0.966 [38]	
Fume-ash-based geopolymer	0.77 [39]	
Foamed geopolymer	0.092–0.157 [40] 0.079–0.766 [41] 0.030 [42] 0.0947–0.01273 [43]	
Ceramic brick	0.56 [44]	
Reinforced concrete	1.7 [44]	

## Data Availability

All data are contained within the article.

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
