# Peer review of "The Effect of Mineral Wool Fiber Additive on Several Mechanical Properties and Thermal Conductivity in Geopolymer Binder"

_materials, 2024, doi:10.3390/ma17020483_

Round 1
Reviewer 1 Report
Comments and Suggestions for Authors
The authors need the revision of the manuscript for publication in Materials journal. Some questions and suggestions are as follows;
1. We strongly believe that it is necessary to systematically study the mechanical properties and coefficient of thermal conductivity when the contents of the stone wool and glass wool are varied. The discussions for this version to achieve the research results in these fields are weak. Therefore, the authors need the observation of optimum content and systematic formulation study to present the scientific information.
2. We suggest that Authors should show the further chemical composition and chemical bonding information of the samples using X-ray photoelectron spectroscopy (XPS) and/or Fourier transform infrared spectroscopy (FT-IR) to increase understanding of this manuscript by journal readers.
3. We suggest that Authors should add the related literature about Figure 1.
4. We suggest that authors should add the scale bar in several figures.
5. We believe that authors can merge and/or delete some figures in Figures 2, 3, 4, and 5.
6. We suggest that authors should change the detailed figure caption, especially Figure captions 2 and 8.
7. This manuscript has too many typographic errors. We suggest that authors should correct typographic errors such as superscript and/or subscript issues and spelling checks (strenght vs. strength and decimal point vs. decimal comma) in the whole manuscript.
Comments on the Quality of English LanguageModerate editing of English language is required.
Author Response
We would like to thank the Reviewer for this review. Please find attached our tesponces to inniwidual comments.

Reviewer 2 Report
Comments and Suggestions for Authors
This paper describes a method of preparing geopolymer from mineral wool waste. Two samples were prepared by adding glass wool and asbestos. The compressive strength, bending strength and thermal conductivity of the two samples were measured. The results show that the addition of fiber significantly improves the bending strength, but does not improve the insulation properties of the material itself, which has certain research value.
However, there are the following problems, please modify:
1. The abstract of this paper is too complicated and verbose, not organized, can not clearly express the research content of this paper, it is suggested to rewrite;
2. There are no quantifiable results in the abstract, which are basically qualitative conclusions, which cannot convince readers, please revise;
3. Please add the last paragraph in the Introduction to explain the differences between this paper and previous literature studies, so as to reflect the value of this study, please add;
4. The conclusion part of this paper does not discuss the research conclusions of this paper, and it is suggested to rewrite them.
5. Part 4 of the paper, which discusses the conclusions of mechanical properties and thermal conductivity, is too simple. Please elaborate on the underlying reasons based on the analysis results.
Author Response
We would to thank the Reviewer for this review. Please find attachment our responces to individual comments.

Reviewer 3 Report
Comments and Suggestions for Authors
This manuscript discusses the impact of the addition of waste mineral wool fibers to the geopolymer binder. The paper determines the properties of the geopolymer after the addition of fibers such as compressive and flexural strength and thermal conductivity coefficient. The manuscript verify whether the addition of mineral wool fibers has a beneficial effect on the properties of the geopolymer. The results have proven that the addition of fibers significantly improves flexural strength.
This is an article of interest. However, the authors must defend the novelty of their research and how it differs from other similar works published in the literature.
Furthermore, the description of section 2.2, which is very brief, should be improved.
Author Response
We would to like to thank the Rewiewer for this review. Please find attachment our responces to individual comments.

Round 2
Reviewer 1 Report
Comments and Suggestions for Authors
-
Comments on the Quality of English Language-